# The Application of DTCWT on MRI-Derived Radiomics for Differentiation of Glioblastoma and Solitary Brain Metastases

**DOI:** 10.3390/jpm12081276

**Published:** 2022-08-03

**Authors:** Wen-Feng Wu, Chia-Wei Shen, Kuan-Ming Lai, Yi-Jen Chen, Eugene C. Lin, Chien-Chin Chen

**Affiliations:** 1Department of Radiology, Ditmanson Medical Foundation Chia-Yi Christian Hospital, Chiayi 600, Taiwan; cych06454@gmail.com (W.-F.W.); 13196@cych.org.tw (K.-M.L.); 2Department of Chemistry and Biochemistry, National Chung Cheng University, Chiayi 621, Taiwan; jasper060@gmail.com (C.-W.S.); wwhhaattt8@gmail.com (Y.-J.C.); 3Department of Medical Imaging and Radiological Sciences, Central Taiwan University of Science and Technology, Taichung 406, Taiwan; 4Department of Pathology, Ditmanson Medical Foundation Chia-Yi Christian Hospital, Chiayi 600, Taiwan; 5Department of Cosmetic Science, Chia Nan University of Pharmacy and Science, Tainan 717, Taiwan; 6Department of Biotechnology and Bioindustry Sciences, College of Bioscience and Biotechnology, National Cheng Kung University, Tainan 701, Taiwan

**Keywords:** artificial intelligence, dual-tree complex wavelet transform, glioblastoma multiforme, machine learning, magnetic resonance imaging, solitary brain metastasis, wavelet transform

## Abstract

Background: While magnetic resonance imaging (MRI) is the imaging modality of choice for the evaluation of patients with brain tumors, it may still be challenging to differentiate glioblastoma multiforme (GBM) from solitary brain metastasis (SBM) due to their similar imaging features. This study aimed to evaluate the features extracted of dual-tree complex wavelet transform (DTCWT) from routine MRI protocol for preoperative differentiation of glioblastoma (GBM) and solitary brain metastasis (SBM). Methods: A total of 51 patients were recruited, including 27 GBM and 24 SBM patients. Their contrast-enhanced T1-weighted images (CET1WIs), T2 fluid-attenuated inversion recovery (T2FLAIR) images, diffusion-weighted images (DWIs), and apparent diffusion coefficient (ADC) images were employed in this study. The statistical features of the pre-transformed images and the decomposed images of the wavelet transform and DTCWT were utilized to distinguish between GBM and SBM. Results: The support vector machine (SVM) showed that DTCWT images have a better accuracy (82.35%), sensitivity (77.78%), specificity (87.50%), and the area under the curve of the receiver operating characteristic curve (AUC) (89.20%) than the pre-transformed and conventional wavelet transform images. By incorporating DTCWT and pre-transformed images, the accuracy (86.27%), sensitivity (81.48%), specificity (91.67%), and AUC (93.06%) were further improved. Conclusions: Our studies suggest that the features extracted from the DTCWT images can potentially improve the differentiation between GBM and SBM.

## 1. Introduction

Glioblastoma multiforme (GBM) and brain metastasis are the most commonly identified brain neoplasms in the adult population [1,2]. The distinction between these two types of tumors is critical for future therapeutic planning. These two diseases have quite different treatment strategies. For example, maximum safe resection followed by concomitant chemoradiotherapy needs to be considered for GBM [3], while stereotactic radiosurgery or en bloc resection can be performed for brain metastases [4]. Accurate diagnosis with advanced imaging modalities provides information about these two tumors without needing to obtain histopathologic evidence, posing the risk of morbidity and mortality. Magnetic resonance imaging (MRI) is generally recommended as the imaging modality in clinical practice for differentiating GBM and brain metastasis.

GBM typically presents with central necrosis surrounded by an enhancing rim of tumor cells with peritumoral edema [5]. Brain metastasis tends to be located between the gray–white matter interface [6], solitary or multiple. A cerebellar position, multifocal dissemination, clear boundaries, and clinical information on systemic cancer favor the diagnosis of brain metastasis over glioblastoma in patients with enhancing brain masses [7,8]. However, brain metastases can present as a solitary (named SBM in the following context) lesion approximately half of the time [6,9], which may cause difficulty in differentiation with GBM on conventional and advanced MRI due to insufficient and variable validity of these imaging modalities [10].

The radiomics-based extraction of mineable high-dimensional data from MRI for texture analysis can provide the underlying information related to pathophysiology, which is arduous to detect by visual perception [11,12]. The extracted features are helpful as a complementary tool to the professional human reader. Therefore, the images should be treated as high-dimensional data [13]. The goal of precision medicine could be to aid with machine learning and improve diagnosis, prognosis, and prediction in clinical practice [14]. Radiomics was first used in oncology research, but it can potentially be used for any disease [13], including neurologic, thoracic, genital–urinary, breast, gastrointestinal, hematologic, and musculoskeletal radiology [15,16]. The typical pipeline of radiomics predominantly involves the following steps: image acquisition, segmentation, image preprocessing, feature extraction, dimension reduction, model validation, and performance evaluation [11,12,17].

As mentioned, the boundaries of GBM and SBM would have different characteristics in the MR images, suggesting that algorithms sensitive to the edges would benefit from the distinction, such as the wavelet transforms mentioned in the online (version 5) image biomarker standardization initiative (IBSI) reference manual [18]. However, the conventional wavelet transforms only emphasize vertical and horizontal features of the images that conflict with the nature of tumors which have round or irregular margins. This could be why little attention has been paid to incorporating wavelet transformation into the workflow of radiomics for differentiation of GBM and SBM [19,20,21,22,23,24,25]; however, it has been suggested that incorporating it could improve the performance of differentiation [19]. To avoid the restraints from the conventional wavelet transforms, we applied dual-tree complex wavelet transformation (DTCWT) [26] to address this problem, which deciphers the image details by six orientations (±15°, ±45°, and ±75°). Some studies have shown that Alzheimer’s disease and multiple sclerosis (MS) can be distinguished from normal brains using DTCWT [27,28,29]; however, to the best of our knowledge, there are no reports on the application of DTCWT to brain tumors.

Several advanced MRI techniques, such as diffusion imaging, perfusion imaging, spectroscopy, and diffusion tensor imaging, have been used to discriminate between the two entities [30,31,32]. Likewise, the discriminations could be improved with enhanced sensitivity and resolution at a stronger B_0_ field. However, these techniques are not conducted routinely or universally owing to long acquisition times, inconsistent accuracy, or high cost. Conventional MRI techniques at a 1.5 T system are still primarily used in clinical practice. Herein, we attempted to differentiate GBM and SBM from the MRI images with routine protocols acquired at 1.5 T. We focused on the features obtained from the wavelet transformations, especially DTCWT, which could reveal more structure information.

## 2. Materials and Methods

### 2.1. The Patient Enrollment

We collected the cases of GBM and brain metastasis from 1995 to 2020. There were 62 pathology-proven GBM cases found in the GBM database. Of the 62 GBM cases, 27 cases were recruited and retrospectively entered into this study. On the other hand, 85 pathology-proven cases of brain metastases were found in the brain metastasis database. Of the 85 cases with brain metastases, 24 cases with SBM were recruited and retrospectively entered into this study. In total, 51 patients were recruited between 2007 and 2020 (27 GBM and 24 SBM) in this retrospective analysis which the local Institutional Review Board approved (the approval number: IRB2021051 and the approval date: 31 May 2021). For the 24 patients with SBM, histopathological analysis revealed 13 cases that originated from the lung, five cases from the breast, two from the liver, one from the kidney, two from the colon, and one from the ovary. Inclusion criteria were as follows: (1) patients of single GBM or SBM with pathologically confirmed according to the WHO criteria [33]; (2) available preoperative MRI with multiparametric protocol; (3) newly diagnosed cases without history of treatment. The exclusion criteria were as follows: (1) examinations with artifacts or missing sequences; (2) extra-parenchymal masses; (3) subjects with multiple lesions.

### 2.2. MRI Research Protocol

The preoperative MRI was collected from the picture archiving and communication systems (PACS) at Ditmanson Medical Foundation Chia-Yi Christian Hospital. In detail, 19 GBM and 14 SBM cases were performed on a 1.5 T Signa™ HDxt scanner (GE Healthcare, Milwaukee, WI, USA) with an eight-channel neurovascular array GE coil and fast spin echo (FSE) sequence. Eight GBM and 10 SBM cases were performed on a 1.5 T Optima™ MR450w (GE Healthcare, Milwaukee, WI, USA) scanner with a 16-channel GE head-and-neck unit coil and the PROPELLER (Periodically Rotated Overlapping Parallel Lines with Enhanced Reconstruction) sequence. The duration of each complete examination was about 20 min with ear plugs.

The acquisition protocols of the MRI examinations included FSE T1-weighted image (T1WI) (repetition time/echo time (TR/TE), 2600/24 ms; rephasing radiofrequency (RF) pulse, 160°; section thickness, 6 mm, intersection gap, 0.6 mm; number of acquired signals, 1; matrix, 320 × 224; field of view (FOV), 220 mm × 220 mm), FSE T2 fluid-attenuated inversion recovery (T2FLAIR) (TR/TE, 9000/140 ms; inversion time (TI): 2200 ms; rephasing RF pulse, 160°; section thickness, 6 mm, intersection gap, 0.6 mm; number of acquired signals, 1; matrix, 320 × 224; FOV, 220 mm × 220 mm), single-shot echo planar imaging (SS-EPI) diffusion-weighted image (DWI) (TR/TE, 8000/76.6; section thickness, 6 mm; intersection gap, 0.6 mm; number of acquired signals, 1 for b value 0 s/mm^2^ and 2 for b value 1000 s/mm^2^; matrix, 128 × 128; FOV, 240 mm × 240 mm; b values, 0 and 1000 s/mm^2^), and contrast-enhanced T1-weighted image (CET1WI). We routinely apply FSE and fat suppression (FS) techniques on CET1WI. After a check-up of the patients’ renal function, CET1WI was obtained with the T1WI sequence about 30 s after intravenous administration of a standard dose of gadobutrol (Gadovist^®^, 0.1 mmol/kg body weight). The array spatial sensitivity encoding technique (ASSET) was applied for DWI sequences to accelerate image acquisitions. The apparent diffusion coefficient (ADC) maps were generated automatically using the GE built-in algorithm. All the images acquired were two-dimensional with static MRI scans.

### 2.3. Imaging Processing and Analysis

The operational flow of the imaging processing and analysis is shown in Figure 1. A radiologist (W.-F.W.) and a radiographer (K.-M.L.) manually selected a region of interest (ROI) from all of the CET1WI images independently. These ROIs were selected once; hence, no intra-observer analysis was available. The intersection of selected ROIs from an image was utilized in the subsequent studies. T2FLAIR, DWI, and ADC images were aligned to the corresponding CET1WI images. The pixels out of the skull region were treated as the background and removed, and then the images were normalized using a histogram with 100 bins [34]. These images then could be transformed using discrete wavelet transform (DWT) with the Haar wavelet or DTCWT. Combining the imaging types (CET1WI, T2FLAIR, DWI, and ADC) and transformed approaches (six in DTCWT, four in DWT, and the pre-transformed image), 44 images could be derived from a patient. The ROIs of these images were extracted using six statistical metrics: mean, coefficient of variation (CV), skewness, kurtosis, energy, and entropy. The definition of these metrics is listed in the Appendix A. This extracted information was then selected using a *t*-test. Finally, we used the linear support vector machine (SVM) to distinguish GBM and SBM on the basis of the possible combination of the selected features [19,20], in which the SVM was performed with a fivefold cross-validation (the ratio of training and validation data was 4) and regularization parameter, c, of 1. All the analyses were performed using MATLAB^®^ 2021a (The MathWorks, Inc., Natick, MA, USA).

## 3. Results

The representative CET1WI and T2FLAIR images and the selected ROIs are shown in Figure 2. No apparent signatures could be used to distinguish between the GBM and SBM images. The corresponding DWI and ADC images are shown in Appendix A. The CET1WI and T2FLAIR images were further transformed using DWT or DTCWT (Figure 3). The approximation component of DWT indicates a tumor profile; hence, these images lose the details and vary less in amplitude.

In contrast, these details are depicted in DWT’s horizontal, vertical, and diagonal components and six orientations in DTCWT. The corresponding wavelet-transformed DWI and ADC images are shown in Appendix A. In this particular pair of GBM and SBM patients, the wavelet-transformed images seem to have more differences between the GBM and SBM patients, suggesting the potential of using wavelets to distinguish GBM and SBM patients.

By applying the wavelet transforms and statistical features to these images, 264 features could be generated from a patient. Then, we used a *t*-test to select 21 features with a *p*-value smaller than 0.001 between GBM and SBM for further investigation (as shown in Table 1). For the pre-transformed images, the six selected features were all from the CET1WI images. For the DWT images, most of the statistical features from the CET1WI approximation images were selected, and the kurtosis of the T2FLAIR diagonal image was selected. For the DTCWT images, the selected features were mainly the ±15° and ±45° of T2FLAIR images with skewness, kurtosis, and entropy. No features from ADC and DWI images were selected on the basis of the *p*-value.

However, by examining the box plots of these selected features (Appendix A), the distributions of a feature from GBM and SBM still could overlap, which indicates the challenge of differentiating GBM and SBM on the basis of a sole feature. Hence, features from the pre-transformed, DWT, and DTCWT images were further grouped to distinguish between GBM and SBM using SVM. Specifically, we considered all possible combinations of features from the pre-transformed (63 combinations, i.e., 2^number of features^−1 = 2^6^−1), DWT (63 combinations), and DTCWT (511 combinations) images. We also considered all the combinations (2^21^−1) of the selected features from the pre-transformed, DWT, and DTCWT images.

The best feature combination from each group is shown in Table 2. DTCWT had the highest accuracy (82.35%), followed by the DWT (76.47%) and pre-transformed (72.55%) images. Likewise, DTCWT had the highest area under the curve (AUC) of the receiver operating characteristic (ROC) curve (89.20%), followed by the DWT (88.89%) and pre-transformed (84.26%) images. DTCWT also had the highest specificity (87.50%) and sensitivity (77.78%). When all the possible combinations were considered, SVM selected three and six features from the pre-transformed CET1WI and DTCWT T2FLAIR images, respectively. The accuracy (86.27%), sensitivity (81.48%), specificity (91.67%), and AUC (93.06%) were further improved. We also included another statistical analysis, F1-score, to evaluate the classification, which showed a similar result to AUC.

## 4. Discussion

Several features observed on conventional MRI (e.g., T1WI, CET1WI, T2WI, and T2FLAIR) [5], such as tumor morphology, distribution, number, enhancing pattern, and peritumoral T2 prolongation, have been used to differentiate between GBM and brain metastasis [35,36]. However, these morphologic characters are nonspecific and prone to high interobserver variability. Advanced MRI (e.g., diffusion-based techniques) [5] utilizing multiple parameters to assess cellular density, microvascular permeation, vascular proliferation, and tissue metabolites improves the diagnostic performance of classifying these two disease entities in comparison with conventional MRI [20,37,38,39]; nonetheless, the advanced MRI protocols are sensitive to the acquired and analytic methods. In addition, several other studies demonstrated inconsistent discrimination results [31,40,41,42]. However, the discrepancies among the studies make it difficult for advanced MRI to guide clinical practice.

The radiomics analysis quantifies the information of texture features extracted from MRI, enhancing clinicians’ existing data. Currently, radiomics-based analysis forces a shift from the visual perception of medical images, which is highly variable, to the extraction of highly dimensional meaningful data to support clinical decision making for precision medicine [11,12,13,14,15,16,17]. In addition, several reports utilizing texture features for differentiation between high-grade gliomas or GBM and brain metastases demonstrated AUC results in the range between 0.68 and 0.96 [19,20,23,43,44].

An image can be transformed into frequency domains by employing a wavelet series. These wavelets are the images’ high- or low-pass filters and result in the approximation, vertical, horizontal, and diagonal components. The approximation component can be further transformed, resulting in a hierarchical multiresolution image [45,46,47]. With these unique properties, wavelet transformation has been applied in many fields, including medical imaging [48]. However, these wavelets were proposed to capture the horizontal or vertical details, which may not be suitable for detecting the physiological morphologies. DTCWT extracts the image features from six different orientations, including ±15°, ±45°, and ±75°. Compared with DWT, DTCWT offers a higher degree of directional selectivity [49,50,51], allowing more information on the morphologic features of an image with increased robustness in the orientations. Therefore, it is suggested that the energies of the decompositions from the DTCWT identify the difference between MS patients and healthy volunteers [27]. In the present study, we proposed applying the DTCWT technique on radiomics-based machine learning with routine MRI sequences to discriminate patients with a single GBM from those with SBM.

Among the 264 radiomics features obtained from a patient, 21 features were selected by *t*-test with a *p*-value <0.001 (Table 1). We also attempted to select the features using principal component analysis (PCA). However, we found that (1) none of the principal components (PC) dominates (<20%), and (2) the feature weightings in each PC are small (e.g., <10% in PC1). The feature selection using PCA only provided ambiguous results. The selected features based on a *t*-test were all associated with CET1WI or T2FLAIR images, and none of the DWI or ADC features were selected. Our results are consistent with the previous finding that using ADC values for quantitative analysis does not help differentiate GBM and metastasis [41,42]. The presumed causes could be that DWI or ADC has a lower spatial resolution, signal-to-noise ratio, and contrast-to-noise ratio [52]. Even though the diffusion-based images seemed ineffective in this study, we think that the potential of the diffusion images could also have been diminished by our ROI selection (based on CET1WI). The lesions from the diffusion images might still be valuable with a proper choice of ROI [53].

The selected features of the pre-transformed images (Table 1) were all from CET1WI, which might reflect the solid enhancing components of GBM and SBM [5]. However, it also pointed out that using the pre-transformed images’ statistical features might only have limited accuracies. The features selected from DWT images were mostly the approximation component, which suggests the tumor profiles would provide more information than the morphology details (e.g., vertical, horizontal, and diagonal components) to distinguish between GBM and SBM. Likewise, the morphology details from CET1WI images transformed by DTCWT also showed little difference. However, the features of T2FLAIR seemed potent to DTCWT (orienting on ±15° and ±45°), which suggests that the heterogeneity differences between GBM and SBM could be identified. The differences in morphology details are not apparent enough to be detected by DWT and DTCWT on post-contrast images (CET1WI). Gadolinium-based contrast medium shortens the T1 relaxation rates of water proton [54] to increase the T1 imaging contrast. The contrast agents diffusing into interstitial spaces enhance the tissue contrast between normal and diseased structures [55]. However, the contrast agents could also diminish the slight difference in the relaxations within a tumor. Herein, the loss of the morphology details resulting from the contrast agents might explain that DWT and DTCWT detected no useful features from the details of the CET1WI images. In contrast, these useful features of the morphology details from T2FLAIR images extracted using DTCWT could indicate the fine-structure differences between GBM and SBM.

There were several limitations to this study. Firstly, radiomics studies are based on retrospectively collected data and tend to have varied imaging protocols, attenuating the reproducibility of radiomics features and classifiers [17]. It is crucial that radiomics studies are compliant with the IBSI guidelines [56] to ensure reproducibility and validity of the outcomes. The compliance assessment of our radiomic software program for the IBSI standard needs further investigation. Herein, we listed the items we followed from the IBSI checklist in Appendix A. Secondly, only 51 patients (27 GBM and 24 SBM) were included in the study due to the limited size of our database. Although we employed fivefold cross-validation to minimize the overfitting due to the small sample size, it still requires additional test data to verify its generalizability. Thirdly, the ROIs’ intra- and interobserver variability might be introduced due to the manual segmentation, which could be minimized by utilizing automatic or semi-automatic algorithms, which are potentially more reproducible [11]. Fourthly, further investigations of the tumor cell infiltration and edema (i.e., the peritumoral region) might improve the differentiation based on the DWI and ADC images [5,57]. However, our extracted features were segmented from the enhancing and non-enhancing necrotic areas of the tumor without including the peritumoral region. Consequently, our segmentations based on the CET1WI images might have resulted in the ineffectiveness of diffusion images for differentiating GBM and SBM. Further work is required to employ the concept of tumor habitat imaging in radiomics for potential valuable disease-specific cues [58]. Fifthly, we only employed the basic wavelet, Haar, as the benchmark of the conventional wavelet, which could slightly underestimate the performance of DWT. However, our results showed that the useful features for the differentiation of GBM and SBM were from CET1WI for DWT and T2FLAIR for DTCWT, which indicated that DTCWT and the conventional DWT identified the different image characteristics. Hence, the underestimated performance of DWT using Haar wavelet might have been negligible. Lastly, employing the wavelet features to differentiate GBM and SBM was based on the assumption that these two kinds of tumors had specific information from different directions. However, we might have missed the links between these directional features and histology characteristics.

## 5. Conclusions

To our knowledge, there have been no reports about the application of DTCWT for the differentiation of GBM and SBM. This application of DTCWT demonstrated the technical feasibility in feature extraction and dimensional reduction of an image for distinguishing GBM and SBM with high performance. Significantly, DTCWT improves the ability to analyze the details of extracted images with six different orientations, which enormously increases the robustness of orientations and relevantly conforms to the actual tumor morphologies while doing radiomics-based machine learning. In conclusion, the application of DTCWT on routine MRI in differentiating GBM from SBM with the approach of radiomics-based machine learning is feasible with a favorable and comparable (comparative) diagnostic accuracy compared with the performances of DWT and pre-transformed images. Further study with a more significant amount of sample data and inclusion of the peritumoral region for texture analysis is required to improve diagnostic accuracy and expand generalizability.

## Figures and Tables

**Figure 1 jpm-12-01276-f001:**
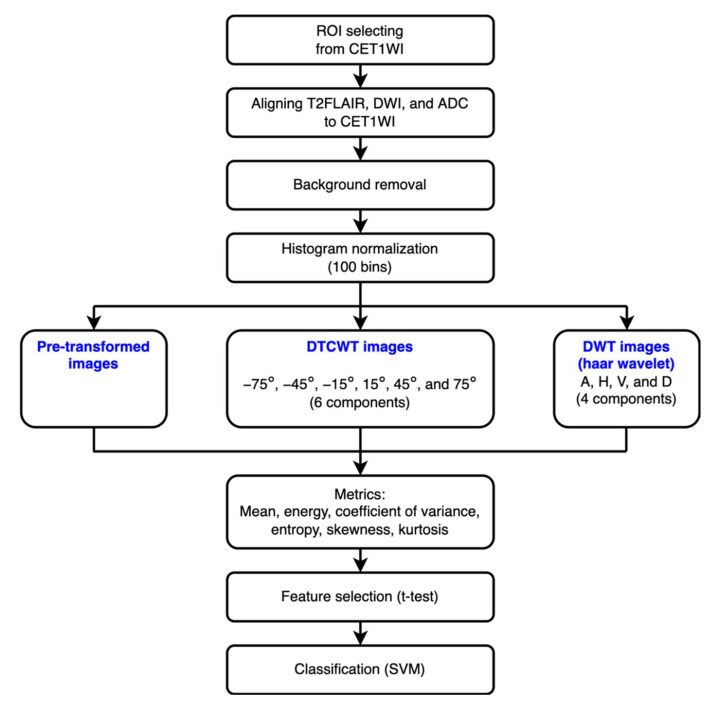
The flowchart of imaging processing and distinguishing. Briefly, a region of interest (ROI) was selected on the basis of fast spin echo (FSE) contrast-enhanced T1-weighted imaging (CET1WI) with fat suppression (FS) images. Then, T2 fluid-attenuated inversion recovery (T2FLAIR) image, diffusion-weighted image (DWI), and apparent diffusion coefficient (ADC) were aligned to the corresponding CWT1WI image. The backgrounds of these images were then removed, and the images were then normalized before the discrete wavelet transform (DWT) and dual-tree complex wavelet transform (DTCWT) and analysis. Finally, the features were selected using a *t*-test, and glioblastoma multiforme (GBM) and solitary brain metastasis (SBM) were differentiated using the support vector machine (SVM).

**Figure 2 jpm-12-01276-f002:**
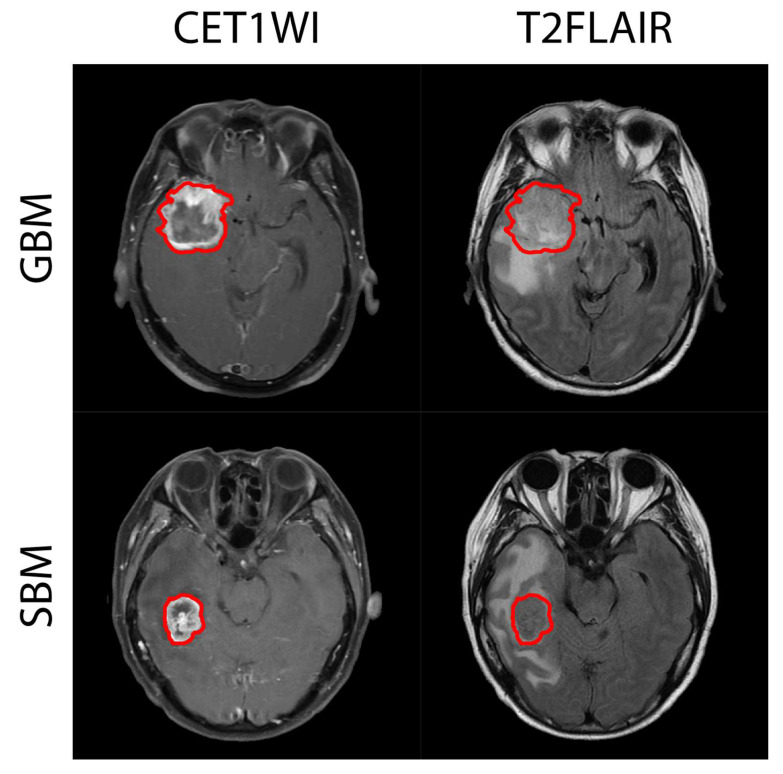
The representative fast spin echo (FSE) contrast-enhanced T1-weighted imaging (CET1WI) with fat suppression (FS) images and T2 fluid-attenuated inversion recovery (T2FLAIR) images from the glioblastoma multiforme (GBM) and solitary brain metastasis (SBM) patients. The enclosed regions with the red boundary were utilized for the analysis.

**Figure 3 jpm-12-01276-f003:**
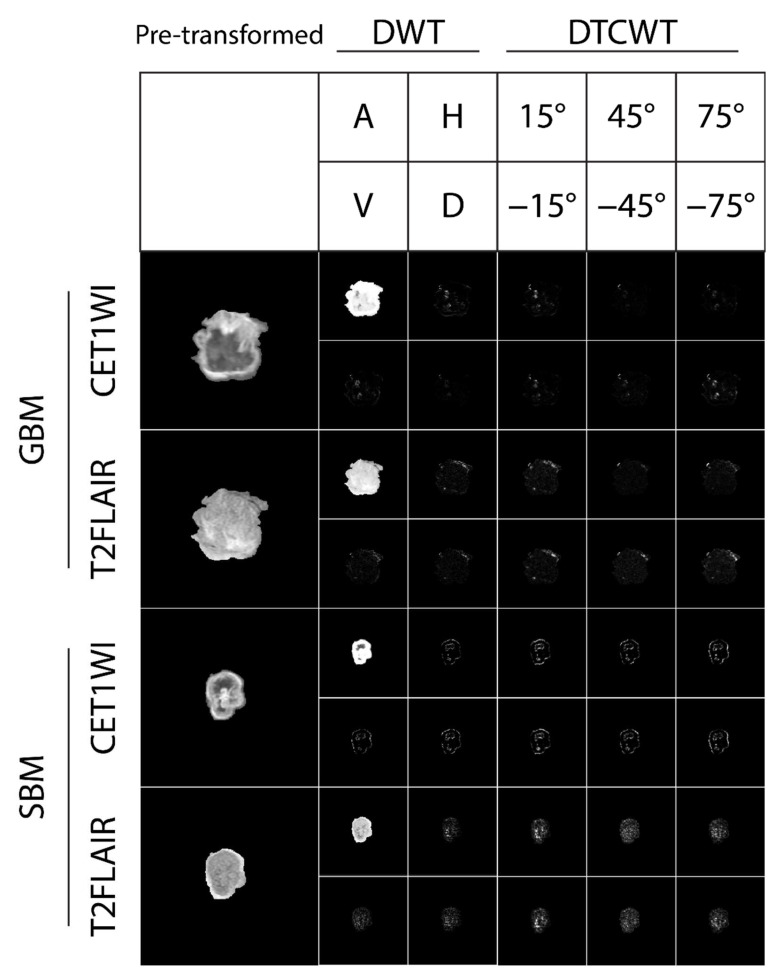
The enlarged region of interests (ROIs) from Figure 2 and the corresponding wavelet-transformed images. The top panels show the arrangement of the wavelet-transformed images. DWT, discrete wavelet transform; DTCWT, dual-tree complex wavelet transform; A, approximation; H, horizontal; V, vertical; D, diagonal.

**Table 1 jpm-12-01276-t001:** The selected features using a *t*-test with a criterion of *p* < 0.001.

Pre-Transformed		DWT			DTCWT		
Image	Feature	Image	COMP	Feature	Image	COMP	Feature
T1	Mean	T1	A	Mean	T2	15°	Skewness
T1	Energy	T1	A	Energy	T2	15°	Entropy
T1	CV	T1	A	Skewness	T2	45°	Skewness
T1	Skewness	T1	A	Kurtosis	T2	45°	Kurtosis
T1	Kurtosis	T1	A	Entropy	T2	45°	Entropy
T1	Entropy	T2	D	Kurtosis	T2	−45°	Skewness
					T2	−15°	Skewness
					T2	−15°	Kurtosis
					T2	−15°	Entropy

COMP, the component of wavelet decomposition; DWT, discrete wavelet transform; DTCWT, dual-tree complex wavelet transform; CV, coefficient of variation; A, approximation; D, diagonal; T1, CET1WI; T2, T2FLAIR.

**Table 2 jpm-12-01276-t002:** The best results of distinguishing between GBM and SBM based on the selected features of the pre-transformed, DWT, and DTCWT images.

Performance	Pre-Transformed	DWT	DTCWT	Pre-Transformed + DWT + DTCWT
ACC (%) (CI)	72.55 (59.05–82.89)	76.47 (63.24–85.99)	82.35 (69.74–90.43)	86.27 (74.27–93.19)
SEN (%) (CI)	74.07 (54.13–87.36)	70.37 (50.37–84.75)	77.78 (58.04–89.86)	81.48 (62.07–92.21)
SPC (%) (CI)	70.83 (51.98–84.49)	83.33 (65.38–92.97)	87.50 (70.26–95.40)	91.67 (75.43–97.53)
AUC (%) (CI)	84.26 (71.93–91.79)	88.89 (77.41–94.92)	89.20 (77.79–95.12)	93.06 (77.25–98.15)
F1-score	74.07%	76.00%	82.35%	86.27%
Image/(COMP)/Feature				
	T1/energy T1/skewness	T1/A/energy T1/A/entropy T2/D/kurtosis	T2/45°/skewness T2/45°/kurtosis T2/−15°/skewness	**Pre-transformed** T1/kurtosis T1/mean T1/skewness
				**DTCWT** T2/15°/skewness T2/45°/kurtosis T2/45°/skewness T2/−45°/skewness T2/−15°/entropy T2/−15°/kurtosis

GBM, glioblastoma multiforme; SBM, solitary brain metastasis; DWT, discrete wavelet transform; DTCWT, dual-tree complex wavelet transform; ACC, accuracy; SEN, sensitivity; SPC, specificity; AUC, the area under the curve of the receiver operating characteristic (ROC) curve; CI, confidence interval; T1, CET1WI; T2, T2FLAIR

## Data Availability

Data are available on request due to all institutional restrictions related to patient privacy.

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
