# Peer review of "The Application of DTCWT on MRI-Derived Radiomics for Differentiation of Glioblastoma and Solitary Brain Metastases"

_jpm, 2022, doi:10.3390/jpm12081276_

Round 1

Reviewer 1 Report

This paper deals with a topic of interest: the limits of conventional imaging to make a differential diagnosis between the most common primary brain tumor, i.e., glioblastoma, and the brain neoplasia more frequent in adults, i.e., the brain metastasis. The authors propose a radiomics model. The paper appears well written, and results presented are interesting. However, a supporting statistical analysis is lacking. Particularly, I would ask the author to support data of table 2 with an appropriate statistical analysis.

Regarding the MR technique described in the materials and methods, are the parameters, the protocol, and the scanner the same for all patients? This is an important point as the cases have been enrolled since 1995. The authors should describe in more detail the type of T1 sequences after gadolinium. Are those turbo spin echo images? From Figure 1, they appear images with pulse for fat signal suppression; Please, provide adequate information.

Please check the sentence, lines 108 and 109.

Author Response

Dear Editor in Chief and my respectful reviewers

            We are really grateful for your expertise and sincere comments on our manuscript. All your comments are right to the point and we have revised the manuscript according to your recommendations.

Reviewer 1:

Comments to the Author

  1. This paper deals with a topic of interest: the limits of conventional imaging to make a differential diagnosis between the most common primary brain tumor, i.e., glioblastoma, and the brain neoplasia more frequent in adults, i.e., the brain metastasis. The authors propose a radiomics model. The paper appears well written, and results presented are interesting. However, a supporting statistical analysis is lacking. Particularly, I would ask the author to support data of table 2 with an appropriate statistical analysis.

-> We included another statistical analysis, F1-score, to evaluate the classification in Table 2, which shows a similar result to AUC.  

  1. Regarding the MR technique described in the materials and methods, are the parameters, the protocol, and the scanner the same for all patients? This is an important point as the cases have been enrolled since 1995. The authors should describe in more detail the type of T1 sequences after gadolinium. Are those turbo spin echo images? From Figure 1, they appear images with pulse for fat signal suppression; Please, provide adequate information.

-> We thank the reviewer pointed out our negligence in the detailed description of the MRI parameters, protocols, and scanners. The MR images of 19 GBM and 14 SBM patients were obtained using a 1.5 T Signa HDxt scanner (GE Healthcare, Milwaukee, USA) with an FSE sequence. Those of 8 GBM and 10 SBM patients were obtained using a 1.5 T Optima MR450w scanner (GE Healthcare, Milwaukee, USA) with the PROPELLER sequence. We routinely apply fast spin echo and fat suppression techniques to the CET1WI sequences. The image of CETIWI in Figure 1 was acquired with FSE and fat suppression. We have added this detailed information to the caption of Figure 1.

  1. Please check the sentence, lines 108 and 109.

->We added the details of the sequence employed in CETIWI.

Reviewer 2 Report

This is an interesting study that employed a relatively new aspect of machine learning in the distinction between GBM and SBM with encouraging results. A few aspects need to be considered and addressed before considering this work for publication:

-          The sample size of the current study is very small, and it lacks a validation (neither internal or external validation was performed). Therefore, the provided results are of uncertain applicability and reproducibility.

-          There are some minor language and typo issues that need to be revised.

Examples: “sensitive” in the abstract, “The region of interests” in page 3 of 13 line 123, “with where selected” in page 6 of 13 line 173, “it still requires additional test data to verify its generality” in the discussion page 9 of 13 line 283 etc. Please also polish the whole manuscript for any overlooked minor languages issues.

-          Only a total of 51 patients have been recruited over a span of 25 years of relatively common diseases. Please clarify the mode of data collection: random, successive, or others? what could explain this limited data over this long time span?

-          Were all patients scanned by the same MRI scanner with fixed parameters over 25 years? We know that MR imaging techniques have significantly changed and advanced in the past three decades.

-          Two readers have selected the ROIs, was this independent? Different parts of the dataset by each? or they segmented the same dataset twice? Any inter- and intra-observer analysis performed?

-          What is the definition of “the as-acquired images”? the pre-processed?

-          “The region of interests (ROIs) were selected based on the CET1WI images, and then CET1WI, DWI, and ADC images were aligned to the corresponding T2FLAIR images”. If the ROI was initially selected on CET1WI, then all other sequences should be aligned to this selection from CET1WI. I don’t understand how these sequences were then aligned to FLAIR. This statement is confusing, please revise .. Additionally, it’s obvious from the representative images within the main manuscript and in the supplementary materials that copying the ROIs from one sequence to the other is not the best way to delineate the lesions on different sequences. This ended up including bony structures and other irrelevant tissues into the selection and analysis. It would have been better if you have considered each sequence as an independent imaging occasion with independent selection and feature calculation. Eventually, combinations may be considered directed by the statistical analysis results.

-          I liked that you have emphasized the importance of adherence of the radiomics studies to the IBSI initiative guidelines, but can you expand your statement to acknowledge the limitations of your study in terms of adherence to this guideline.

-          The limitations section, overall, needs to be revised to speak about the limitations specific to this current work rather than broadly speaking about the limitations associated with radiomics studies in general.

Author Response

Dear Editor in Chief and my respectful reviewers

            We are really grateful for your expertise and sincere comments on our manuscript. All your comments are right to the point and we have revised the manuscript according to your recommendations.

Reviewer 2:

Comments to the Author

This is an interesting study that employed a relatively new aspect of machine learning in the distinction between GBM and SBM with encouraging results. A few aspects need to be considered and addressed before considering this work for publication:

  1. The sample size of the current study is very small, and it lacks a validation (neither internal or external validation was performed). Therefore, the provided results are of uncertain applicability and reproducibility.

Thank you for your precise comments. We have mentioned this limitation in the last paragraph of the discussion and suggest further investigation and validation with large-scaled case series or multi-institutional research.

  1. There are some minor language and typo issues that need to be revised.

Thank you for the careful comments. We have thoroughly checked the spelling and grammar with commercial software.

  1. Examples: “sensitive” in the abstract, “The region of interests” in page 3 of 13 line 123, “with where selected” in page 6 of 13 line 173, “it still requires additional test data to verify its generality” in the discussion page 9 of 13 line 283 etc. Please also polish the whole manuscript for any overlooked minor languages issues.

Thank you for the careful comments. We have revised these language issues according to your comments and thoroughly checked the spelling and grammar with commercial software.

  1. Only a total of 51 patients have been recruited over a span of 25 years of relatively common diseases. Please clarify the mode of data collection: random, successive, or others? what could explain this limited data over this long time span?

Thank you for your precise comments. Our hospital has established the databases of GBM and brain metastases since 1995, and this study includes cases collected from 1995 to 2020. There were 62 pathology-proved GBM cases, and 27 of them were recruited retrospectively in this study according to the inclusion and exclusion criteria. On the other hand, there were 85 pathology-proved brain metastases cases, and 24 of them were recruited retrospectively in this study according to the inclusion and exclusion criteria. The earliest enrolled GBM and SBM patients were in 2007 and 2009, respectively. The exclusions include but are not limited to no available MR images before the operation, sequence missing, motion artifacts, multifocal tumors, or a combination of the above. Therefore, we have revised the timespan to avoid confusion.

  1. Were all patients scanned by the same MRI scanner with fixed parameters over 25 years? We know that MR imaging techniques have significantly changed and advanced in the past three decades.

We thank the reviewer pointed out our negligence in the detailed description of the MRI parameters, protocols, and scanners. The MR images of 19 GBM and 14 SBM patients were obtained using a 1.5 T Signa HDxt scanner (GE Healthcare, Milwaukee, USA) with an FSE sequence. Those of 8 GBM and 10 SBM patients were obtained using a 1.5 T Optima MR450w scanner (GE Healthcare, Milwaukee, USA) with the PROPELLER sequence. In addition, we routinely apply fast spin echo and fat suppression techniques to the CET1WI sequences. These scanners have no hardware or software upgraded since 2007 (as mentioned in response 4).

  1. Two readers have selected the ROIs, was this independent? Different parts of the dataset by each? or they segmented the same dataset twice? Any inter- and intra-observer analysis performed?

Both the two readers separately and independently performed ROI segmentation of the same whole parts of the MR images with the manual method from the enrolled GBM and SBM cases. These two readers only select the ROI once; hence, there is no intra-observer analysis. Instead, we took the intersection of the ROIs from the two readers for the feature extraction; therefore, we did not have an inter-observer analysis. To make it clear, we included this description in the context.

  1. What is the definition of “the as-acquired images”? the pre-processed?

We have changed “the as-acquired” to “pre-transformed” to avoid confusing descriptions.

  1. “The region of interests (ROIs) were selected based on the CET1WI images, and then CET1WI, DWI, and ADC images were aligned to the corresponding T2FLAIR images”. If the ROI was initially selected on CET1WI, then all other sequences should be aligned to this selection from CET1WI. I don’t understand how these sequences were then aligned to FLAIR. This statement is confusing, please revise .. Additionally, it’s obvious from the representative images within the main manuscript and in the supplementary materials that copying the ROIs from one sequence to the other is not the best way to delineate the lesions on different sequences. This ended up including bony structures and other irrelevant tissues into the selection and analysis. It would have been better if you have considered each sequence as an independent imaging occasion with independent selection and feature calculation. Eventually, combinations may be considered directed by the statistical analysis results.

We are embarrassed that we misplaced CET1WI and T2FLAIR in the description of alignment, as the reviewer pointed out. We have rephrased the sentence and the relating content.  
We agree with the review’s suggestion that the independent sequence selection and feature calculation prevent contamination from other sequences. In our hospital, the DWI sequence equipped with a single-shot echo planner imaging (SS-EPI) is the most commonly used readout method in the current clinical practice due to its low sensitivity to motion-induced phase errors. However, SS-EPI has limitations of geometric distortions, strong susceptibility artifacts, and poor spatial resolution due to a shorter T2* relaxation time. Our DWI protocol also included the parallel imaging method, array spatial sensitivity encoding technique (ASSET), to improve spatial resolution and lessen susceptibility artifacts. Nevertheless, there is still geometric distortion and blurring in our DWI images, making it difficult to delineate the margin of the tumors during segmentation and align accurately when co-registration. Hence, we used the ROI selected from CET1WI in this study. We also mentioned this potential problem in the limitation.

  1. I liked that you have emphasized the importance of adherence of the radiomics studies to the IBSI initiative guidelines, but can you expand your statement to acknowledge the limitations of your study in terms of adherence to this guideline.

We have listed the items that we followed and did not follow from the IBSI guideline in Table S2 as our supplementary data.

  1. The limitations section, overall, needs to be revised to speak about the limitations specific to this current work rather than broadly speaking about the limitations associated with radiomics studies in general.

We have described more limitations specific to the wavelet transformation in the last paragraph of the discussion.